# Analysis of Selected Properties of Polymer Mixtures Derived from Virgin and Re-Granulated PP with Glass Fibers

**DOI:** 10.3390/ma17061433

**Published:** 2024-03-21

**Authors:** Tomasz Stachowiak, Dariusz Kwiatkowski, Marcin Chmielarz, Dominik Grzesiczak

**Affiliations:** 1Department of Technology and Automation, Faculty of Mechanical Engineering and Computer Science, Częstochowa University of Technology, Armii Krajowej 21 St., 42-201 Częstochowa, Poland; dariusz.kwiatkowski@pcz.pl; 2Granulat-Bis Company, Hallera 8A St., 42-202 Częstochowa, Poland; m.chmielarz@granulatbis.com.pl (M.C.); dominikg88@gmail.com (D.G.)

**Keywords:** recycling, polymer materials, polymer composites, polymer reprocessing

## Abstract

The problem of the growing amount of waste polymer materials currently affects virtually every area of the global economy. New actions taken by the E.U. and member states could lead to a reduction in the burden on the natural environment, as well as the reuse of thermoplastic waste. The aim of this study was to analyze the possibility of reusing post-industrial waste (recycled polypropylene—rPP) in order to produce mixtures with original polypropylene (PP) and glass fibers. The research undertaken is characterized by a high level of innovation and was carried out on an industrial scale from industrial waste. The primary goal of the analyses was to determine changes in the properties of the polymer mixtures depending on the amount of recycled polymers. For this purpose, four types of mixtures were prepared, characterized by different degrees of filling with recycled material obtained from big-bag packaging (the filling levels were 0 wt.%, 20 wt.%, 30 wt.%, and 70 wt.%). A detailed analysis of the physical properties of the obtained mixtures was carried out to determine changes in the densities depending on the amount of rPP material. In addition, changes in the MFIs (melt flow indexes), characterizing viscosity changes, were analyzed depending on the amount of secondary raw material used. An analysis of the mechanical properties was also carried out based on static tensile testing, the impact strength (the Charpy method), and the Rockwell hardness test (the M method). The analysis of the thermal changes was performed using the DSC method. The results showed that the composites made of virgin polypropylene (PP GF30) and those made from re-granulates and glass fibers (rPP GF30) are characterized by similar mechanical properties and significantly different processing properties, determined by MFI. This means that the addition of re-granulates significantly affects the processability of the obtained materials, while the addition of glass fibers maintains the basic mechanical properties.

## 1. Introduction

Year after year, the production of thermoplastics registers a continuous increase. After the stagnation of and significant deterioration in international markets related to the pandemic period, there has been a significant global rebound, leading to a sharp increase in the use and adaptation of polymer materials in new applications. Plastics are used on a large scale in individual markets such as the packaging industry, consumer electronics and household appliances, construction, the electronics and electrical industry, the automotive industry, agriculture and horticulture, medicine, and sports. These are mainly polymeric materials belonging to the group of thermoplastics. Their great advantage is the possibility of reprocessing them, which people often overlook, resulting in them ending up in landfills [1,2,3,4,5,6].

Market analysis indicates that, in 2023, 400 Mt of thermoplastics was produced around the world. Of this, only 9% of recycled plastics and 2.3% of polymer plastics from renewable sources were in circulation worldwide. Only 0.1% of the 400 Mt of plastic produced was chemically recycled and reused to produce polymers [7].

As part of the production of thermoplastics, over 45% of materials produced belong to the polyolefin group (including PE-LD, PE-HD, and PP), of which 18.9% is polypropylene and its composites [7].

The European market has an annual production of thermoplastics of 58.7 Mt, with an 18.5% share of mechanically recycled materials. This value consists of a 12.9% share of post-consumer mechanical recycling and a 5.6% share of pre-consumer mechanical recycling. The European market also includes chemical recycling at the level of 0.2%. The use of bio-based plastics is at the level of 1%, which produces 1.7 Mt per year of bio-based and biodegradable polymers [7].

This low level of the reuse of plastic waste is further hampered by the significant amount of thermoplastic-based composites in circulation. These are mainly composites filled with various types of fibers, mainly glass fibers (so-called short and long fibers and, in a growing number of applications, continuous fibers). In the literature, many examples can be found of the analysis of the impact of recycling or of the use of recycled materials to obtain new polymer materials [4,8,9,10,11,12,13,14,15,16]. The innovative approach in this publication is the analysis of the influence of adding recyclate from pre-consumer waste streams, with the aim of obtaining a mixture of the original polymer with re-granulates and glass fibers not previously subjected to recycling. The mixtures were produced using a compounding extruder (see diagram). Most authors present research results based on the recycling of polymer composites filled with glass fibers or the analysis of the properties of thermoplastic composites made from secondary raw materials [1,2,5,6,9,12,17,18,19,20].

Evens et al. (2019) analyzed the process of manufacturing and recycling composites based on polypropylene filled with 20% glass, carbon, and natural fibers (flax). Selected properties of the obtained standardized samples were analyzed, then they were milled and doped with the original polymer, and the resulting mixture was extruded again. The properties of the newly obtained samples were also analyzed. The authors summarized their research with conclusions indicating the deterioration in the mechanical properties, which, however, depended on the number of recycling rounds and the type of fiber used [21]. Ephraim et al. (2015) analyzed the mechanical properties of glass fiber-reinforced polypropylene re-granulates. The work used an extruder to obtain the re-granulates (unlike the research conducted by Evans), and the waste used was post-consumer waste. The authors used three levels of glass fiber filling: 35, 40, and 50%, and the main goals of their work were to determine the mechanical properties of glass fiber-reinforced plastic based on recycled resin and to assess its suitability for construction purposes. The analysis made it possible to observe differences resulting from the amount of filler and its influence on the obtained mechanical properties [22]. One of the very interesting studies is the analysis conducted by Kuram et al. (2014). The influence of not only the filler in the form of glass fibers but also the addition of polyethylene and its impact on selected mechanical and thermal properties was analyzed [23].

A similar problem was analyzed by El Hajj et al. (2020). The analysis focused on polymer mixtures comprising polymers such as polyamide, polypropylene, and polyethylene derived from the recycling of food-grade cast films. The properties of the materials obtained in this way were augmented with short glass fibers. The primary goals of the work carried out by the researchers were to obtain a homogeneous mixture of the above-mentioned polymers and then produce composites from them. The conducted research allowed for the verification of the properties of the obtained mixtures and newly created materials and the possibility of their reuse. The introduction of short glass fibers into the mixture had a positive effect on the mechanical properties but led to a significant deterioration in the impact strength [15]. In terms of polymer mixtures and the reuse of various types of polymer materials, research was also conducted by Chiang et al. (2020). The work was aimed at obtaining a homogeneous mixture of PP with recycled elastomers using maleic anhydride as a compatibilizer. Comparative analysis allowed for a demonstration of the significant differences in the properties of mixtures and composites in which elastomers were used. The improvement in their thermal stabilities and mechanical properties was mainly influenced by the elastomeric matrix used. The analysis clearly indicates the possibility of using elastomeric waste in future polymer applications based on broadly understood recycling. Intermediary substances and fillers, including glass fibers, significantly improved the selected properties [24].

Achukwu et al. (2023) focused on the analysis of the properties of PP composites filled with glass fibers in the amount of 10 wt.% with repeated processing (the composites were processed five times). The polypropylene for the testing came from in-house waste plastics. The tests showed a deterioration in all the analyzed properties (the physical, mechanical, structural, rheological, and thermal properties of the reprocessed in-house waste) after each reprocessing cycle. The authors pointed out that, for industrial waste, only two reprocessing cycles are necessary, and that, for a larger number of reprocessing cycles, it is necessary to dose the virgin polymer material [1]. The amount of glass fibers in the composite and the processing rate were also analyzed by Achukwu et al. (2022). In their research, they carried out the ten-fold processing of PP composites filled with glass fibers in amounts ranging from 5% to 40%. The obtained results showed that after three cycles of reprocessing, the polymer mixture should be refreshed with positive virgin polymers. Moreover, it was shown that repeated processing (10 times) does not destroy the functional groups present in the polypropylene (based on FTIR). Progressive thermo-mechanical degradation was also demonstrated, resulting in a reduction in the molecular weight [8].

Research in a similar area was also conducted by Colucci et al. Their research focused on analyzing changes in the properties of polymer mixtures consisting of 50 wt.% and 100 wt.% recycled polypropylene and 40% glass fibers. The results obtained by the researchers showed a deterioration in the mechanical properties; however, as the share of recycled material increased, the strain at break increased. The researchers found that reprocessing does not affect the values of the degree of crystallinity, melting point, or degradation temperature (determined using DSC and TGA methods). The researchers conclude that the reprocessing of PP GF40 composites allows for their reuse in structural elements in the automotive industry [2].

One of the equally interesting and innovative directions is the use of polypropylene re-granulates with glass fibers in 3D printing technology. This topic was discussed, among others, by Sam-Daliri et al. (2020 and 2023). The use of recycled materials in applications aimed at using cheap waste raw materials to produce composites using 3D printing was analyzed. As the obtained results show, this is also a very promising direction for the reuse of waste from thermoplastic composites filled with various types of fibers [3,25].

As the research shows, the recycling of polypropylene composites filled with glass fibers is a commonly used solution aimed at reusing waste from many industries. The properties of the material after the recycling process will depend mainly on the length of the glass fibers as well as on the number of times they are processed (Achukwu et al. (2022 and Achukwu et al. (2023) [1,8]). Researchers also point out that in order to obtain or maintain the assumed properties of recycled thermoplastic polymers, it is necessary to dose the original polymers into the re-granulates. It should also be noted that the work so far has focused on the reprocessing of PP composites filled with glass fibers, which means that, during reprocessing, the fibers are shortened as a result of the mechanical impact (grinding) and reprocessing (extrusion and/or injection). Our research focused on an innovative approach to the issue of recycling. The aim of our research was to determine the impact of mixing Moplen HP500N polypropylene with re-granulates (from the industrial waste stream) and virgin glass fibers (in the amount of 30 wt.%) on the processing and functional properties of the finished product. It should be emphasized that neither the virgin polypropylene nor re-granulates contained glass fibers before the compounding process (before the mixing process and creation of the polymer composition). Moreover, the conducted research showed that adding virgin material to the mixture does not improve the obtained properties (contrary to the conclusions drawn by Achukwu et al. (2022) and Achukwu et al. (2023)). The obtained results allowed for the verification of the influence of the re-granulates and their quantity as well as the addition of glass fibers on the properties of the obtained composites. The analysis showed the influence of the re-granulates and 30 wt.% glass fibers on the processability of the mixtures (defined by changes in the mass flow rate index), as well as on the mechanical properties, impact strengths, and thermal properties.

## 2. Materials and Methods

Two types of polymer materials and glass fibers were used for the tests. The first group of materials included Moplen HP500N virgin polypropylene (more information is provided in Table 1) from LyondellBasell Industries. The polymer was supplied by Basell Orlen Polyolefins (Płock, Poland).

The second group of materials used was re-granulated PP (rPP) derived from the recycling of industrial waste from big-bag packaging. The re-granulates was produced and supplied by Granulat-Bis (Czestochowa, Poland). The material (rPP) was produced using a recoSTAR dynamic 85C-VAC extruder (Recyclingmaschine—STARLINGER & CO Ges. m. b. H., Vienna, Austria). The polymer obtained in the extrusion–granulation process was characterized by a melt flow rate of 6 g/10 min. The analysis of the melt flow rate index of the rPP material was carried out in accordance with the ISO 1133-1 standard, using the LMI 500 capillary plastometer Dynisco (Heilbronn, Germany). The melt flow rate tests were performed by the Granulat-Bis company as part of the development of the properties of polypropylene re-granulates (rPP).

TGFS 202P glass fibers with a length of 4 mm and fiber tensile strength of 2000 MPa, supplied by Taiwan Glass, were used as a filler.

The aim of the research was to produce polymer composites based on polypropylene filled with 30 wt.% glass fibers. The produced composites were characterized by the following proportions (Table 1). The amount of recycled material was selected based on the literature as well as E.U. guidelines, which, in the latest directives, specify the amount of recycled material that should be used as an additive in new products (this value ranges from 20 to 30 wt.%). Moreover, many studies state that the addition of re-granulates at a level of up to 10 wt.% does not significantly affect the properties of the new material, but it also does not contribute to the increase in the use of recycled materials. An amount of re-granulates above 40 wt.% may contribute to the deterioration in the processability of the mixture and the properties of the finished product (mechanical properties, thermal properties, impact resistance, etc.). Excessive amounts of re-granulates may also lead to the formation of a significant amount of gases as a result of the re-granulate degradation process. Similar dependencies were noticed for glass fibers; therefore, in order to obtain the optimal mechanical properties, composites with a filling of 30 wt.% were produced. In addition, one variable was analyzed, which was the changing amount of re-granulates in the composite.

The first material consisted of 70 wt.% virgin polypropylene Moplen HP500N and 30 wt.% glass fibers (material code: PPGF30).

The second material was produced as a mixture containing 50 wt.% Moplen HP500N polypropylene, 20 wt.% polypropylene re-granulates (rPP) (obtained from big-bag waste), and 30 wt.% glass fibers (material code: PPGF30-50/20).

The third material was produced as a mixture containing 40 wt.% Moplen HP500N polypropylene, 30 wt.% polypropylene re-granulates (rPP) (produced from big-bag waste), and 30 wt.% glass fibers (material code: PPGF30-40/30).

The last material was produced as a mixture containing 70 wt.% re-granulated polypropylene (rPP) (produced from big-bag waste) and 30 wt.% glass fibers (material code: rPPGF30).

This composition was prepared using a compounding extruder (STEER MEGA-58 twin-screw extruder from STEER ENGINEERING LTD, Karnataka, India). The glass fibers were fed directly into the plasticization system of the extruder.

The amounts of re-granulates used were dictated by the minimum noticeable re-granulate value in the case of the 50/20 composition, while the 40/30 composition was the maximum re-granulate value that could be used without a significant change in or impact on the properties of the finished product, as shown by the melt flow rate tests, the processing properties deteriorate as the share of re-granulate increases (the melt flow rate decreased significantly). Further, increasing the amount of re-granulates could significantly worsen the processability of the composition.

The input material for the composites was original PP with a melt mass flow rate of 12 g/10 min. The dosed re-granulates had a melt flow rate of 6 g/10 min (measurements of the mass flow rates of the materials were made in accordance with the ISO 1133-1 standard, using plastometer LMI 500—Dynisco). It should be noted that the above-described melt flow index measurements were performed for the individual substrates included in each polymer composition (Moplen HP500N and rPP). The following part of the publication contains the results of the MFI values for the obtained mixtures filled with glass fibers (30 wt.%).

### 2.1. Tested Materials

In response to the increasing demand for polymeric materials and their composites, as well as the growing issue of waste from polymer materials, and considering the concept of the circular economy, an analysis was conducted to verify the changes in the properties of polypropylene-based polymer composites depending on the content of re-granulates derived from the recycling process. As already mentioned above, polymer mixtures were prepared for testing and then subjected to the compounding process using a STEER MEGA-58 twin-screw extruder from STEER ENGINEERING LTD. Table 2 shows the shares of the individual components of the PP/rPP/glass fiber composite and the basic processing properties of the individual materials included in the composition. Unfilled plastics were mixed, and 30 wt.% glass fibers were added during the extrusion process (in the final stage of the compounding process).

### 2.2. Technologies Used to Manufacture Granulates and Standardized Parts

The technological parameters for the injection and extrusion processes were determined by analyzing technological cards (TDSs), literature data, and information on the extrusion and injection process provided by the Granulat-Bis company (Poland), concerning the processing of re-granulates and materials obtained during the compounding process. Despite the different contents of secondary material in the form of re-granulates from the reprocessing of big-bag waste, the same processing parameters were used for all mixtures during the injection and extrusion. This was due to the desire to avoid the additional impact of the process parameters on the properties of the obtained mixtures (composites). In recent years, for economic and ecological reasons, the process parameters have decreased significantly. In industrial conditions for PP, the injection or extrusion temperature values (for standardized samples) oscillate around 230 °C. Significant modifications of the processing parameters have been introduced in relation to complex geometries (e.g., with thin walls and long flow paths and/or MFI fluctuations) [26,27,28].

Glass fiber-modified polymer mixtures were obtained in the extrusion–granulation process. A STEER MEGA-58 twin-screw extruder from STEER ENGINEERING LTD was used. The mixing of individual substrates was carried out to achieve full homogenization. The following processing parameters were applied: the extruder die temperature was set at 190 °C, the plasticization system temperature (zone before the extrusion die) was maintained at 215 °C, and the filler zone was set at 20 °C.

After leaving the extrusion die, individual polymer threads were immersed in a water bath at a temperature of 30 °C. Two cooling baths were used with a total cooling path length of 5.2 m. After cooling the granulate and removing excess moisture, the polymer threads were cut using a granulator. Figure 1 shows the scheme for obtaining the tested mixtures using the compounding extruder STEER MEGA-58.

Standardized test samples were produced from the obtained re-granulates using the injection method. The samples were made in accordance with the PN-EN ISO 294-1 standard: Plastics—Injection moulding of test specimens of thermoplastic materials—Part 1: General principles, and moulding of multipurpose and bar test specimens. They were produced using the following injection parameters: melt temperature: 230 °C; injection pressure: 450 bar; injection speed: 75 mm/s; clamping pressure: 230 bar; mold temperature: 40 °C.

### 2.3. Analysis of Density Differences of Obtained Granulates

Due to the introduction of original polypropylene and recycled materials into the process, one of the fundamental tests carried out was density analysis. It was carried out for samples taken from standardized moldings (the density of the obtained granulates was not analyzed). Density measurement was carried out in accordance with the ISO 1183-1 standard: Plastics—Methods for determining the density of non-cellular plastics—Part 1: Immersion method, liquid pycnometer method and titration method [29].

One of the first tests carried out for the polymer mixtures characterized by different degrees of filling with recycled polymer (different shares of re-granulates) was a comparison of the densities of the obtained composites. The study was carried out to verify the influence of the re-granulates and their contents in the polymer mixtures on the basic physical properties of the manufactured composites. The test was carried out using the immersion method on a measurement station consisting of a Radwag AS 220.3Y laboratory balance with auxiliary equipment.

### 2.4. Analysis of Mechanical Properties—Tensile Test

The analysis of the mechanical properties was performed for all produced polymer materials. Standardized type 1A samples were prepared using injection technology. The samples were made in accordance with the PN-EN ISO 294-1 standard: Plastics—Injection of test shapes from thermoplastics—Part 1: General principles [30], forming standardized samples and beam shapes.

The analysis of the mechanical properties was carried out in accordance with the standard ISO 527-1: Plastics—Determination of tensile properties—Part 1: General principles [31].

The samples were stretched at a speed of 50 mm/min. Three repetitions were made for each type of material. During the measurement, the following values were determined: failure stress, maximum force obtained during the test, and deformation obtained. In order to determine these values, the universal testing machine AGX-V with a 50 kN force cell by the Shimadzu Company (Kyoto, Japan) was used. The samples were successively mounted axially in the measuring holders and then stretched until failure in accordance with a previously prepared program.

The analysis of the static mechanical properties was carried out to determine the impact of the re-granulates and their amounts in the polymer composites on the changes in the longitudinal modulus of elasticity, maximum tensile force, and deformability.

The analyzed materials can be successfully reused in industrial applications, replacing the original materials used so far or reducing their consumption, while having a positive impact on the natural environment and contributing to the reduction in waste. However, in order for this goal to be achieved, it is necessary to verify the properties of these types of mixtures.

### 2.5. Melt Flow Rate (MFR) of Obtained Granulates—Test Procedure

The analysis of the flow rate index was carried out in order to determine the viscosities of the individual components (i.e., the flow rate indexes of the individual granulates included in the mixture were analyzed (they are presented in Section 3.3). Then, after the mixing and extrusion with the granulation process, the obtained granulates were analyzed. The resulting values are the values of the melt mass flow indexes for the individual mixtures with different contents of recycled polymers.

To determine the melt mass flow rate, a Dynisco LMI 500 capillary plastometer was used. MFI measurements were carried out in accordance with the standard ISO 1133-1: “Plastics standard—Determination of the melt mass-flow rate (MFR) and melt volume-flow rate (MVR) of thermoplastics—Part 1: Standard method” [32]. The measurements used a weight of 2.16 kg and a measurement temperature of 275 °C.

Melt mass flow rate analysis was performed to determine the effect of the addition of glass fibers (30 wt.% for each composite), as well as the recycled plastic contents. Before starting the measurement, the samples were dried to remove residual moisture. PP is not a hygroscopic polymer; however, during the recycling or granulation process, small amounts of moisture may be present in the materials, which could affect the measurement. To conduct the analysis of the melt mass flow rate using the plastometer, individual granules weighing 10 g each were measured. Within a single measurement, the melt mass flow rate was determined for 5 measurement sections. The averaged value based on five measurements (from five measurement sections) was provided as the final result of the assessment.

### 2.6. Charpy Impact Strength Analysis of Standardized Test Samples Obtained from Re-Granulates—Notched Samples

In order to determine the changes in the dynamic mechanical properties, impact strength analysis was performed in accordance with the ISO 179-1 standard: Plastics—Determination of Charpy impact properties—Part 1: Non-instrumented impact test [33]. The tests used standard samples with type A notch cuts, 2 mm deep, with an angle of 45° and a rounding radius of 0.25 mm. A Zwick/Roell notch cutter was used to make the notches. The impact strength test was carried out using a Zwick/Roell Hit 5.5P impact strength measuring device.

After cutting the notches, the samples were placed on the supports of the measuring device with a support spacing of 62 mm, and then the hammer lock was released, and the sample was broken. Five repetitions were performed for each sample type.

### 2.7. Analysis of Changes in Hardness of Samples Standardized Using HRM Method

Mechanical analysis was also performed for changes in the hardness. The hardness measurements were carried out to determine the influence of the amount of PP re-granulates on the properties of the composites. The tests were carried out in accordance with the PN-EN ISO 2039-2:2002 standard: Plastics, Determination of hardness, Part 2: Rockwell hardness standard [34]. The M-type measurement method was used in the analysis. To determine the hardness, a steel ball with a diameter of 6.35 mm was used.

To determine the hardness, a SINOWON DigiRock DP3 hardness tester was used. During the measurement, the standardized samples obtained in the injection process were placed successively on the measuring table. Before starting the actual measurement, an initial force of 98.8 N was applied to the samples, followed by a specific force of 980 N. In order to remove viscoelastic effects in the tested samples, the specific force was applied for 15 s. The hardness values obtained for each measurement were read from the device’s display.

### 2.8. Analysis of Thermal Properties Using DSC Method

The analysis of the thermal properties was carried out in accordance with the PN-EN ISO 11357 standard: Plastics—Differential scanning calorimetry (DSC)—Part 1: General principles [35], and a POLYMA 214 differential scanning calorimeter from NETZSCH was used for the analysis. Aluminum measuring crucibles and weighed samples of the tested materials of approximately 10 mg were used. An inert gas atmosphere was used in the measurement cell and, for this purpose, technical nitrogen with a purity of 5.0 was used. The tests were carried out using the H-C-H thermal program (heating–cooling–heating). This thermal program was used to remove the thermal histories of the materials being tested. This publication only presents the results of the second heating.

## 3. Results and Discussion

### 3.1. The Results of the Analysis of the Density Measurements of the Tested Materials

Density measurements were performed for samples taken from previously produced standard samples. Five measurement repetitions were made for each material, and the values presented in the charts are the average values (Figure 2 and Table 3).

As shown by the density measurements of the samples obtained from standardized moldings, the average of this value for a polymer derived from re-granulates containing impurities (in the form of contamination and inclusions) is higher than that for samples made from uncontaminated re-granulates (virgin PP). As the share of re-granulates increased (with a constant amount of glass fibers and the same processing parameters), the densities of the obtained samples increased. However, the observed differences are not significant. The densities of the samples made from re-granulates (material marked as rPP GF30) are 4.5% higher than the densities of the samples made of the material marked as PP GF30. For the analysis of the standard deviation values, it should be noted that the obtained values are repeatable.

### 3.2. The Results of the Analysis of the Mechanical Properties of the Tested Samples

Tensile strength tests were carried out in three replications for each of the tested samples, and then the averages were drawn, which are presented in collective bar graphs. The results were divided into three groups related to the obtained values: the moduli of elasticity, maximum deformations, and maximum stresses of the tested samples [36]. The comparison of the obtained results is presented in Figure 3, Figure 4 and Figure 5. The statistical data are presented in Table 4, Table 5 and Table 6.

An analysis of the mechanical properties using a static tensile test was used to present the differences in the mechanical properties of the obtained polymer mixtures. The highest value of the tensile strength was obtained for the samples made of PP material without re-granulates. The obtained tensile strength value is 10.82% higher than the mixture containing 20 wt.% of PP GF30-50/20 re-granulates. Studies have shown that with the increasing share of re-granulates in the mixture, the mechanical properties deteriorate significantly. Increasing the share of re-granulates from 20% to 30% in the mixture leads to a reduction in the tensile strength by another 4.84% (3439 MPa). It should also be noted that by analyzing the mechanical properties of the samples made only from re-granulates, it was possible to demonstrate that the tensile strength is slightly lower than that of the mixture marked as PP GF30-50/20. This means that adding re-granulates to the original polymer materials does not significantly reduce the tensile strength.

The analysis of the obtained deformation values for the composites allowed us to demonstrate that the highest strain value was observed in the composites made of the original polymer (PP) and re-granulates (rPP). The deformability of the mixtures (i.e., materials marked as PP GF30-50/20 and PP GF30-40/30) is approximately 20% lower than the deformability of the original material. Attention should also be paid to the opposite trend than that observed when comparing the mechanical properties. For the analysis of the strain values, the increasing share of re-granulates had a positive impact on the level of deformation of the tested moldings. Therefore, for the material described as PP GF30-40/30, the obtained percentage deformation value is higher than that obtained for the material labeled PP GF30-50/20.

In summary, the creation of polymer mixtures consisting of the original material and re-granulates had a negative impact on the mechanical properties (including the tensile strength and strain). In the case of standardized parts made from re-granulates, a slight deterioration in the mechanical properties was noted compared to the parts obtained from the original material (PP—Moplen HP500N). The reason for the decrease in the mechanical properties is the increasing share of re-granulates. It should be noted that due to the same amount of glass fibers in all the polymer compositions and their significant amount (30 wt.%), the deformations at break for all the materials are similar (in particular for the PP GF30 and rPP GF30 materials). The addition of glass fibers of a uniform and repeatable length leads to low deformations but strengthens and stabilizes compositions containing larger amounts of re-granulates or that are made entirely of re-granulates. A similar relationship was observed for the longitudinal moduli of elasticity. The material with the highest value was made of original PP filled with 30 wt.% glass fibers. The second highest value of the longitudinal elastic modulus was obtained for the material marked as rPP GF30.

The results indicate that the making of polymer compositions consisting of virgin material and re-granulates may significantly affect the properties of the finished product. This also seems to be confirmed by the analysis of the tensile strength and deformation during the tensile tests. The samples marked as PP GF30 and rPP GF30 are characterized by the highest values. One of the reasons may be the process of preparing and homogenizing the mixtures. Moreover, the values of the mass flow rate indexes differ significantly among the mixed materials. The material marked as PP GF30 was produced on the basis of Moplen HP500N polypropylene (PP MFI = 12 g/10 min), while the rPP was produced on the basis of re-granulates (rPP MFI = 6 g/10 min). The remaining blend composites (PP GF30-50/20 and PP GF30-40/30) are characterized by worse mechanical properties and strain as the re-granulate contents increase. Confirmation of this trend can also be observed in the impact strength, but it does not appear in the hardness measurements.

### 3.3. The Results of the Analysis of the Differences in the Values of the Melt Mass Flow Rates of the Tested Materials

The analysis of the melt mass flow rates was carried out using method A in accordance with the ISO 1133 standard. The melt mass flow rates (MFRs) were determined, and the values were determined in the unit g/10 min. The above-described tests were carried out to determine the impact of the re-granulate contents in the polymer mixtures on the processing and rheological properties of the composites. The results are presented graphically in Figure 6. The statistical data are included in Table 7.

The conducted research clearly showed that with the increasing share of re-granulates in the mixture, the technological parameters defined by the melt mass flow rate index deteriorated significantly. From the initial value of 4.93 g/10 min for the sample labeled PP GF30, the MFR decreased to 2.22 for the granulate labeled rPP GF30. The melt mass flow rate decreased by over 53%. The increasing amount of re-granulates in the PP composites therefore negatively affected the polymers’ fluidity and processability. The decrease in fluidity and the increase in viscosity depend on the properties of the material itself (rPP). The base material for obtaining re-granulates was industrial waste from big-bag packaging. The basic melt flow rate for the granulate obtained from this waste was 6 g/10 min. After the compounding process, it decreased to 2.5 g/10 min. This also allowed us to observe that the creation of polymer mixtures with different MFI values significantly affected the resulting values and processability of these compositions, and adding fillers in significant amounts (30 wt.%) (in this case, glass fibers) significantly increased the viscosities of the mixtures.

### 3.4. The Results of the Analysis of the Differences in the Values of the Impact Properties of the Standardized Samples Obtained from Re-Granulates

The aim of the research was also to compare the dynamic properties. For this purpose, an impact strength analysis was performed. Five repetitions were made for each type of material, and the values presented in the graph are the averages of the five measurements (Figure 7 and Table 8). As shown by the impact strength measurements, the composite obtained as a mixture of original polypropylene with glass fibers had the highest impact resistance. The tests also showed that very good values were obtained for the samples made entirely from re-granulates. These were the second-highest values.

The lowest impact strength values were obtained for the samples consisting of a mixture of the original polymer (PP) with re-granulates (rPP).

The reason for the lower impact strengths in the mixed samples may be the homogenization method or differences in the melt mass flow index values.

This means that the obtained composites based on a mixture of original polypropylene (PP) and re-granulates (rPP) are characterized by worse impact strengths. Similar results were obtained for the analysis of the mechanical properties. The mixtures (PP-rPP) were characterized by lower tensile strengths, lower moduli, and lower strains at break. In future applications, it seems that it will be necessary to use materials with similar processing parameters (MFI), or to use compatibilizers or internal lubricants that influence the homogenization process.

In the context of future applications, this is very important information. The research shows that it is more advantageous to process original materials or re-granulates. The formation of the mixtures significantly affects the deterioration in the impact strength, which may be of decisive importance in industrial applications. It also should be noted that the use of re-granulates filled with glass fibers directly in the compounding process may have a positive impact on properties such as the tensile strength and impact strength. This is a significant difference for the current research, which focuses on the analysis of the properties of reprocessed composites (filled polymers). The aim of the research was also an analysis aimed at determining the impact of modifying re-granulates with glass fibers in the process of their reprocessing. As research has shown, this is a promising direction, but it requires further research on the technological process itself, as well as on the production of these types of composites.

### 3.5. The Results of the Analysis of the Hardness Standardized Parts according to the Rockwell (M) Method 

The hardness analysis using the Rockwell method (with an indenter in the form of a steel ball with a diameter of 6350 mm) showed significant differences in its value. Measurements were performed for three samples with three repetitions for each sample. The results in the form of average values are presented in Figure 8. The polymer composite obtained from the virgin polymer (PP) and glass fibers (30 wt.%) without the addition of re-granulate polymer material shows the highest hardness value (Figure 8 and Table 9).

The tests showed that with an increase in the share of PP re-granulates, there was a significant decrease in the hardness values. The lowest hardness value is characteristic of the PPH composite made from re-granulates and not containing the original polymer (PP). The decrease in hardness compared to the original material is 20%.

The results of the hardness measurements are definitely intriguing. Referring to the obtained results of the mechanical properties and impact strengths, a certain trend can be noticed, which, however, is not reflected in the hardness results. The PP GF30 and rPP GF 30 materials are characterized by the best mechanical properties. However, for the hardness measurements, it can be observed that as the amount of re-granulates in the polymer composition increased, the hardness decreased. The reason for this behavior is not only certainly the growing share of re-granulates, but also their composition. While glass fibers play a significant role when analyzing the mechanical properties, in terms of the impact resistance, the excessive stiffness could lead to a decrease in the impact strength value. As the tests showed, for the material marked as rPP GF30, the impact strength is higher than those for the virgin polymer–re-granulate mixtures. This state of affairs may be caused by the participation of other polymer fractions, constituting impurities and affecting the properties. Calorimetric tests (the next section of this publication) showed that as the amount of re-granulates increases, the peak coming from linear low-density polyethylene (LLDPE) becomes more and more visible. This polymer can improve the impact strength of the rPP GF30 composite and, at the same time, significantly decrease its hardness. The homogenization and mixability of the individual substrates in the polymer composition are also important. However, this topic requires further analysis.

### 3.6. Thermal Analysis Results Obtained Using DSC Methodology

Figure 9 shows an overlay of the DSC thermograms obtained during calorimetric measurements. The presented graphs were obtained during the second heating run, after removing the thermal histories.

Table 10 contains a summary of the most important information regarding the obtained values of the melting temperatures for the tested materials, as well as the melting enthalpies.

The tests showed slight differences in the melting temperatures of the individual composites and significant differences in the melting enthalpies. Moreover, during the measurements, it was shown that the obtained polypropylene-based composites (materials labeled PP GF30-50/20, PP GF30-40/30, rPP GF30) were characterized by the occurrence of an additional melting peak around the temperature of 124 °C, which corresponds to the melting point of linear low-density polyethylene. This therefore indicates that the composites were contaminated with other recycled polymer materials. As shown by the tests of the individual properties, including the tensile strength, deformation, impact strength, and hardness, the share of LLDPE may affect the behavior of the polymer composition. Therefore, not only the main substrates and their quantity and the type of filler but also the impurities present in the re-granulates are important. For the analyzed materials, the presence of LLDPE could have become the main factor in the increase in the impact strength and the significant decrease in the hardness of the rPP GF30 material. Moreover, the presence of polyethylene could have resulted in a decrease in the melt flow rate value (apart from the content of glass fibers, which also increases the viscosity of the mixture).

Reusing polymer materials and recycling them without the application and implementation of a system that allows for the precise segregation and separation of different polymer materials may expose us to the uncontrolled production of polymer mixtures with properties that are difficult to predict.

## 4. Conclusions

The aim of the research was to produce polymer mixtures consisting of virgin polypropylene (Moplen HP500N was used in the research) with re-granulates coming from industrial waste streams (the re-granulates were made from waste big-bag packaging). The obtained mixtures were filled with glass fibers in an amount of 30 wt.%. The composites obtained in this way were analyzed for their mechanical properties, impact strengths, and hardness values, as well as their thermal properties.

The tests carried out and the results obtained showed that the mechanical properties of the original polypropylene filled with 30 wt.% glass fibers and the re-granulate PP (rPP) with 30 wt.% glass fibers do not differ significantly. This may be due to the greater repeatability of the material, as well as the homogeneity, which is because only polymer and glass fibers were used for the composites with the PP GF30 and rPP GF30 materials. The materials PP GF30 -50/20 and PP GF30-40/30 are mixtures of virgin PP (Moplen HP500N) and re-granulates. Differences in the MFI values and differences in the amounts of re-granulates may lead to material heterogeneity and, consequently, reduce the mechanical properties.

Significant differences were noted for the hardness values and flow rates of these materials. It has been shown that for composites using re-granulates, the hardness decreases significantly. A similar relationship was observed for the values of the melt mass flow rate index. The reason for the significant differences in the values of these parameters for the PP GF30 material is the presence of polyethylene (LLDPE), which is a contaminant from the re-granulation process. The presence of LLDPE was recorded by thermal analysis (DSC). However, it should be noted that contamination in the form of LLDPE has a positive effect on the impact strength. The impact strength of the rPP GF30 material is only about 11% lower than that of the material obtained from PP Hostalen HP500N (PP GF30). This is a small difference, considering that the rPP GF30 material was made only from re-granulates filled with glass fibers.

Moreover, the conducted research and analysis allowed us to conclude that the production of PP mixtures with re-granulates (rPP) originating from the reprocessing of plastic waste (in this particular situation, from big-bag packaging) leads to deterioration in a number of the analyzed properties. However, for the tested proportions (50/20 or 40/30), the observed differences were not significant. For the PP GF30-50/20 and PP GF30-40/30 materials, which are mixtures, it was shown that the increasing amount of re-granulates does not significantly affect the analyzed properties.

The obtained results also permit us to conclude that, from the standpoint of the derived mechanical characteristics, impact strength, and deformability of the final product, it is advantageous to produce details from re-granulates rather than produce polymer mixtures comprising the original material and recyclate. The reason for this outcome may be the homogenization process resulting from differences in the flow rate values of the materials included in the mixture (PP and PP re-granulates). As the DSC tests showed, contamination may occur by doping the manufactured composites with other polymeric materials originating from the waste stream.

To reduce the amount of consumed polymeric materials and, consequently, mitigate the negative impact on the natural environment, as well as foster the development of industries related to the recycling of thermoplastic materials, it is advantageous, from both an ecological and economic standpoint, as well as purely practical, to utilize polymer materials obtained through recycling. This approach involves using recycled materials without introducing or modifying their properties through the addition of original polymers. The topics of recycling and creating mixtures of virgin materials and re-granulates, as well as the production of these types of composites, require further work and analyses. They will be the focus of subsequent publications.

## Figures and Tables

**Figure 1 materials-17-01433-f001:**
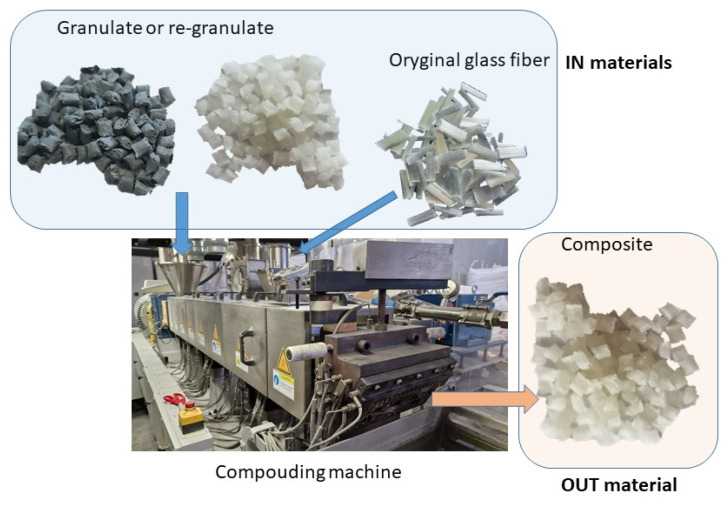
Composite preparation scheme.

**Figure 2 materials-17-01433-f002:**
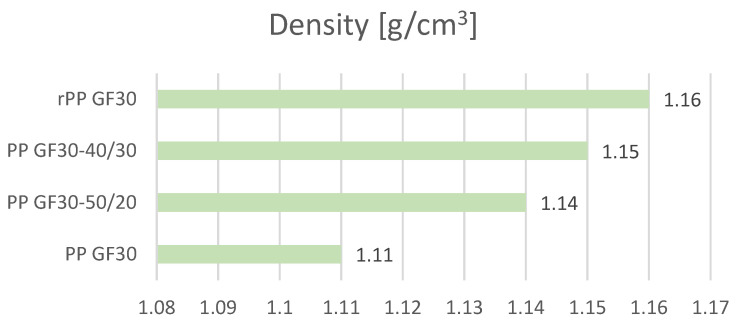
Comparison of the obtained density values.

**Figure 3 materials-17-01433-f003:**
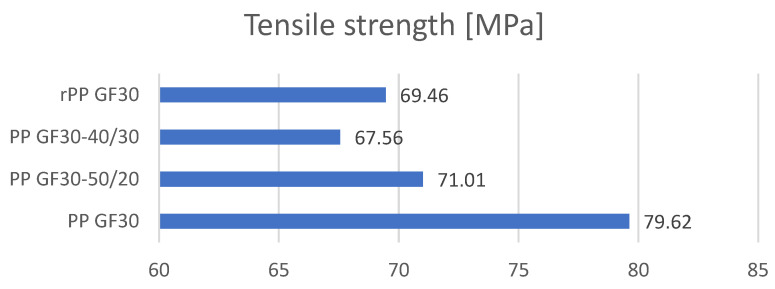
Comparison of the mechanical properties obtained in the static tensile strength test.

**Figure 4 materials-17-01433-f004:**
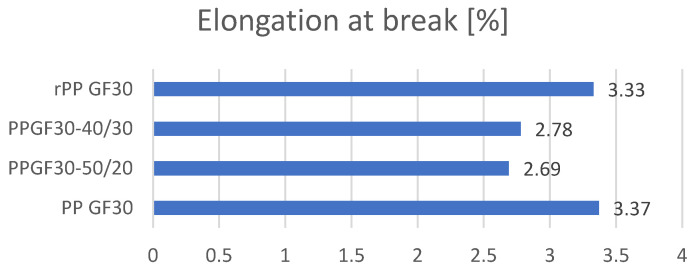
Comparison of elongations at break of obtained composites.

**Figure 5 materials-17-01433-f005:**
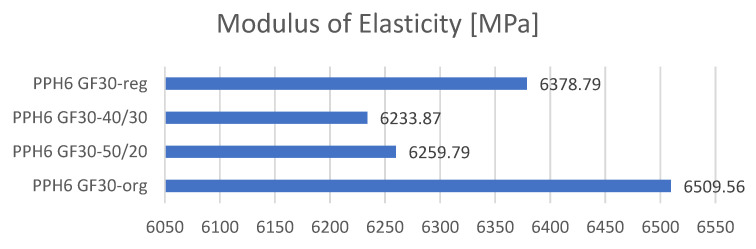
Comparison of moduli of elasticity of obtained composites.

**Figure 6 materials-17-01433-f006:**
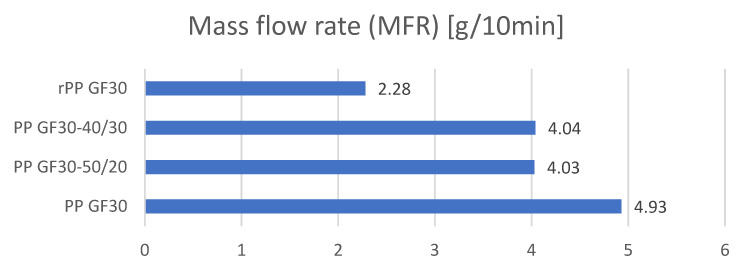
Comparison of the melt mass flow rates of the tested granulates.

**Figure 7 materials-17-01433-f007:**
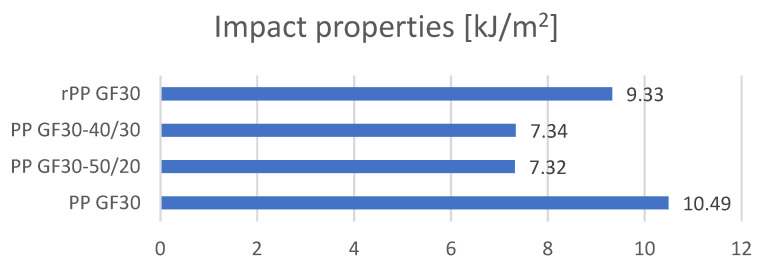
Summary of the results of the impact properties of the tested standardized samples.

**Figure 8 materials-17-01433-f008:**
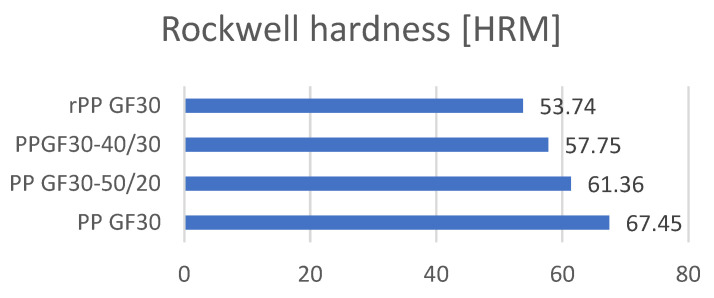
Comparison of the hardness values of the tested samples (HRM method).

**Figure 9 materials-17-01433-f009:**
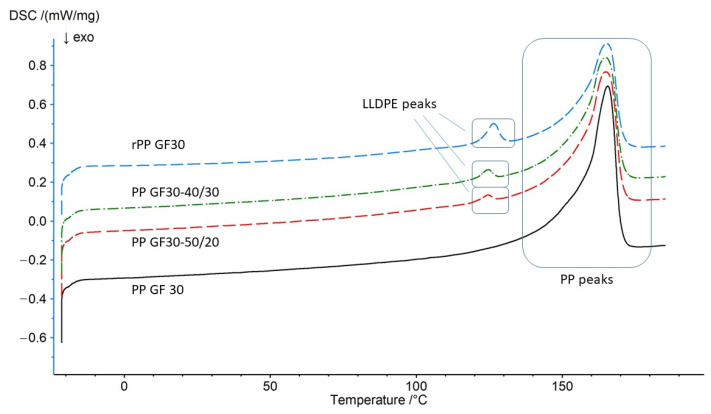
Comparison of the obtained thermograms of the tested granulates.

**Table 1 materials-17-01433-t001:** Formulations of polypropylene/recycled polypropylene/glass fibers.

S./No.	Material Code	PP (Virgin) [wt.%]	rPP [wt.%]	Glass Fibers [wt.%]
1.	PPGF30	70	0	30
2.	PPGF30-50/20	50	20	30
3.	PPGF30-40/30	40	30	30
4.	rPPGF30	0	70	30

**Table 2 materials-17-01433-t002:** Tested materials—specific compositions and mass shares of individual substrates.

PP GF30 Composite				
No.	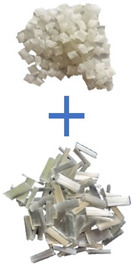	MFR [g/10 min]	Mass [kg]	Amount [wt.%]
PP		12	700	70
rPP		—	0	0
Glass fibers		—	300	30
Output MFR	5 [g/10 min]			
Color	Natural			
**PP GF30-50/20**				
No.	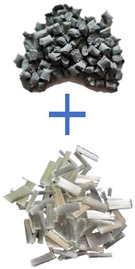	MFR [g/10 min]	Mass [kg]	Amount [wt.%]
PP		12	500	50
rPP		6	200	20
Glass fibers		—	300	30
Output MFR	4.043 [g/10 min]			
Color	Gray			
	**PP GF30-40/30**			
No.	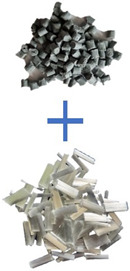	MFI [g/10 min]	Mass [kg]	Amount [wt.%]
PP		12	400	40
rPP		6	300	30
Glass fibers		—	300	30
Output MFR	4.1 [g/10 min]			
Color	Dark gray			
	**rPP GF30**			
No.	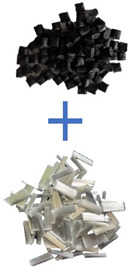	MFI [g/10 min]	Mass [kg]	Amount [wt.%]
PP		—	0	0
rPP		6	700	70
Glass fibers		—	300	30
Output MFR	2.5 [g/10 min]			
Color	Black			

**Table 3 materials-17-01433-t003:** Statistical data regarding density measurements of the obtained composites.

	PP GF30	PP GF30-50/20	PP GF30-40/30	rPP GF30
Avarage value	1.11	1.14	1.15	1.16
Standard deviation	0.0008	0.0011	0.0006	0.0028

**Table 4 materials-17-01433-t004:** Statistical data regarding tensile strength measurement.

	PP GF30	PP GF30-50/20	PP GF30-40/30	rPP GF30
Avarage value	79.62	71.01	67.56	69.46
Standard deviation	0.62	0.87	0.49	0.86

**Table 5 materials-17-01433-t005:** Statistical data—elongations at break of the obtained composites.

	PP GF30	PP GF30-50/20	PP GF30-40/30	rPP GF30
Avarage value	3.37	2.69	2.78	3.33
Standard deviation	0.11	0.01	0.03	0.11

**Table 6 materials-17-01433-t006:** Statistical data—moduli of elasticity of the obtained composites.

	PP GF30	PP GF30-50/20	PP GF30-40/30	rPP GF30
Avarage value	6509.56	6259.79	6233.87	6378.79
Standard deviation	61.84	42.79	179.11	22.16

**Table 7 materials-17-01433-t007:** Statistical data—melt flow rate of obtained composites.

	PP GF30	PP GF30-50/20	PP GF30-40/30	rPP GF30
Avarage value	4.93	4.03	4.04	2.28
Standard deviation	0.12	0.02	0.07	0.03

**Table 8 materials-17-01433-t008:** Statistical data—impact properties.

	PP GF30	PP GF30-50/20	PP GF30-40/30	rPP GF30
Avarage value	10.49	7.32	7.34	9.33
Standard deviation	0.33	0.1	0.18	0.19

**Table 9 materials-17-01433-t009:** Statistical data—Rockwell hardness (M) method.

	PP GF30	PP GF30-50/20	PP GF30-40/30	rPP GF30
Avarage value	67.45	61.36	57.75	53.74
Standard deviation	1.83	1.4	1.41	1.13

**Table 10 materials-17-01433-t010:** Comparison of the obtained values of the melting temperatures and melting enthalpies of the tested materials.

No.	Melting Temp. [°C]	Melting Temp. [°C]	Melting Enthalpy [J/g]
PP GF30-org	-	165.6	77.27
PP GF30-50/20	124.4	164.8	57.75
PP GF30-40/30	124.8	164.7	66.84
PP GF30-reg	126.6	165.2	62.47

## Data Availability

Data are contained within the article.

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
