# Peer review of "Analysis of Selected Properties of Polymer Mixtures Derived from Virgin and Re-Granulated PP with Glass Fibers"

_materials, 2024, doi:10.3390/ma17061433_

Round 1

Reviewer 1 Report

Comments and Suggestions for Authors

Overall, the manuscript presents a comprehensive analysis of the properties of polymer mixtures derived from recycled polypropylene (PP) with 30% glass fiber content. Here are some professional suggestions to enhance the clarity, coherence, and rigor of the manuscript:

Title and Abstract:

The title should be more specific to convey the primary focus of the study.

The abstract should succinctly summarize the purpose, methodology, key findings, and implications of the research.

Introduction:

Provide a clear rationale for the study, explaining why the reuse of polymer waste is important and outlining the specific objectives.

Clarify the significance of the research in addressing current challenges in polymer waste management and sustainability.

Literature Review:

Provide a more structured literature review section that highlights previous research on recycling polymer composites, specifically focusing on PP and glass fiber composites.

Discuss gaps or limitations in existing research that the current study aims to address.

Materials and Methods:

Clearly describe the experimental setup, including details on sample preparation, testing procedures, and equipment used.

Provide justification for the chosen range of recyclate percentages and the methodology for incorporating glass fibers into the composites.

Include information on any standards followed for testing mechanical properties or characterizing the composites.

Results:

Present the experimental results in a structured manner, using tables, graphs, or figures to illustrate key findings.

Include statistical analysis, if applicable, to support the significance of observed differences between different composite formulations.

Discussion:

Interpret the results in the context of the research objectives and previous literature, discussing how the findings contribute to the understanding of polymer recycling and composite material properties.

Address any unexpected or contradictory results and propose possible explanations or avenues for further investigation.

Overall, your manuscript provides a thorough analysis of the properties of polymer mixtures containing regranulates. Here are some key points and suggestions:

1. **Clarity and Structure**: The structure of your manuscript is clear, with sections dedicated to different analyses. This makes it easy for readers to follow the progression of your research.

2. **Standardization and Methodology**: Your use of standardized testing methods, such as ISO standards, adds credibility to your research. It's important to clearly mention the standards used and any deviations from them, if applicable.

3. **Density Analysis**: The comparison of density values provides valuable insights into the effects of impurities and regranulate content on the material properties. However, it would be helpful to discuss the significance of these density differences in relation to practical applications or material performance.

4. **Mechanical Properties Analysis**: Your analysis of tensile strength and deformation properties is comprehensive. Consider discussing the implications of these mechanical property variations on the potential applications of the polymer mixtures.

5. **Melt Mass-Flow Rate (MFR) Analysis**: The decrease in MFR with increasing regranulate content is a significant finding. It would be beneficial to discuss the implications of this decrease on the processability and manufacturability of the polymer mixtures, especially in industrial applications.

6. **Charpy Impact Strength Analysis**: The impact strength analysis provides insights into the dynamic mechanical properties of the polymer mixtures. Consider discussing how variations in impact strength relate to potential applications, such as structural components or packaging materials.

7. **Hardness Analysis**: The analysis of hardness provides additional information about the material properties. Discuss how changes in hardness relate to the structural integrity or durability of the polymer mixtures in different applications.

8. **Thermal Properties Analysis**: Thermal properties analysis using DSC adds valuable information about the material behavior at different temperatures. Consider discussing the implications of variations in thermal properties on the suitability of the polymer mixtures for specific applications, such as thermal insulation or automotive components.

9. **Conclusion**: Your conclusion effectively summarizes the key findings of your research. Consider expanding on the practical implications of your findings and potential future research directions in the field of polymer recycling and composite materials.

10. **Figures and Tables**: Your figures and tables effectively complement the text and provide visual representation of the data. Ensure that all figures and tables are appropriately labeled and referenced in the text for clarity.

Overall, your manuscript provides valuable insights into the properties of polymer mixtures containing regranulates. Consider addressing the suggestions above to enhance the clarity and impact of your research findings.

1. **Impact Properties Analysis**: The comparison of impact properties among different samples provides valuable insights into the dynamic mechanical behavior of the composites. It's important to discuss how variations in impact strength relate to potential applications, especially in terms of structural components or materials subjected to impact loads.

2. **Hardness Analysis**: The significant differences in hardness values among the samples indicate variations in material stiffness and resistance to indentation. Consider discussing how changes in hardness affect the structural integrity or durability of the composites in different applications, such as load-bearing components or surface finishes.

3. **Thermal Analysis**: The DSC analysis reveals important information about the thermal behavior of the composites, including melting temperatures and enthalpies. Discussing the implications of these thermal properties on the processing, performance, and stability of the materials would add value to your conclusions.

4. **Conclusion**: Your conclusion effectively summarizes the key findings of your research. Consider highlighting the practical implications of your findings for industrial applications and future research directions. Additionally, provide a brief recap of the main findings in each section to reinforce the significance of your results.

5. **Discussion of Contamination**: The observation of an additional melting peak in the DSC thermograms suggests contamination with other polymer materials. Discussing the potential sources of contamination and its implications for material properties and processing would enhance the comprehensiveness of your analysis.

6. **Recommendations for Future Research**: Consider including recommendations for future research, such as investigating alternative recycling processes to reduce contamination, optimizing composite formulations to improve mechanical properties, or exploring novel techniques for characterizing material behavior under dynamic loading conditions.

7. **Language and Clarity**: Ensure that the language throughout the manuscript is clear, concise, and free of ambiguity. Clarify any technical terms or concepts that may not be familiar to all readers, and provide sufficient detail to facilitate understanding.

8. **Figures and Tables**: Your figures and tables effectively present the data and results. Make sure that all figures and tables are appropriately labeled and referenced in the text to aid comprehension.

Overall, your manuscript provides valuable insights into the properties and potential applications of polymer composites incorporating recycled materials. Addressing the suggestions above would strengthen the clarity, completeness, and impact of your research findings.

Comments on the Quality of English Language

Moderate editing of English language required

Author Response

Dear Reviewer

At the outset, I would like to thank you for the prepared review. Thank you very much for the extensive study containing such a large amount of valuable information, which will significantly influence the quality not only of the submitted publication but also of future work. I agree with all the comments

Best regards

Tomasz Stachowiak

Reviewer 2 Report

Comments and Suggestions for Authors

·       Remove “30 percent glass fiber content” from the title.

·       In the abstract, shorten lines 11 to 19. In the abstract, you need to summarize what you have done in this work.

·       In the abstract, replace “primary polymer” with “Polypropylene”.

·       Also, in the abstract, rewrite this “Various values of recyclate as a filler were analyzed with a constant value of added glass fiber.” It is not clear what you have done.

·       The abstract must be rewritten and re-organized carefully.

·       The novelty of the work shall be highlighted in the last paragraph of the introduction section.

·       In the last paragraph of the introduction section, what is done, how, and what was found should be presented.

·       What are the reasons for studying these particular composites and what merits do they offer?

·       Where were the materials sourced from? It shall be presented in the section “2. Materials and Methods”.

·       Page 3, line 121, “… using a compounding extruder…”. What was the extruder? Brand? Product name? Country?

·       There are no details regarding the granule-making process.

·       There are no details about the “glass fiber”. Supplier/Manufacturer, length, mechanical properties, etc.

·       The section “2. Materials and Methods” needs to be rewritten. In this section, you must provide essential details for the readers to be able to reproduce the results.

·       What do you mean by “recyclate”??? What is the material?

·       Page 3, line 123, It is not understandable “Then, mixtures containing 20 % of recyclate without fillers in the form of glass fiber (recycled polymer), 30 % and 70 % of recyclate were produced.”.

·       Page 3, line 123, “The recyclate was 124 sourced from the pre-consumer waste stream.”. Where was the waste stream?

·       Page 3, line 128, “The input material for the composites was original PP with a melt mass-flow rate of 128 12 g/10 min. The dosed regranulate had a flow rate of 6 g/10 min.” Did you measure the melt mass-flow rates? If not, add the reference. If so, how??

·       How did you measure the melt mass-flow rates in Table 1? What device? Standard? Testing conditions? etc.

·       In Table 1, had the polypropylene in PP – org and PP – reg the same grade and properties?

·       Table 1 is vague. What is the meaning of the percentages in this table? For example, what is the meaning of 70% for PP – org?

·       What is the meaning of this code “PP GF30 composite – 50/20”? and the similar ones.

·       The paper is not written and organized properly. The manuscript must be re-organized and present a clear description of the experimental work. The first 5 pages are very confusing.

·       Page 5, line 167: Revise this “5,2 meters.”

·       How did you select the processing parameters such as temperatures?

·       Page 6, lines 171-174 are repeated on page 7

·       There are no details regarding the tested samples' dimensions (mechanical tests).

·       The text following the title “2.5. Comparison of melt mass-flow indexes (MFI) of the obtained granulates.” is about the testing procedure not comparison.

·       In the discussion section, you need to discuss the observations not just report them.

·       One of the main manufacturing methods targeting waste plastics is 3D printing with similarities to your work in terms of filament making or pellet making. Add a section in the introduction to include this with similar works to the ones below. In these works, the authors have discussed the reusability and processability of PP reinforced with fibres. In two of them, they exactly used PP+Glass fiber collected from industries. Discuss them in the introduction section.

Recovery of Particle Reinforced Composite 3D Printing Filament from Recycled Industrial Polypropylene and Glass Fibre Waste

Fused Filament Fabricated Polypropylene Composite Reinforced by Aligned Glass Fibers

Experimental study on mechanical properties of material extrusion additive manufactured parts from recycled glass fibre-reinforced polypropylene composite

·       Use bullets to highlight the main achievements of the work in the conclusion section.

Author Response

(The authors gave the same response as above.)

Reviewer 3 Report

Comments and Suggestions for Authors

The authors have presented a study where they investigated properties of recycled polypropylene mixtures with glass fiber filler. The motivation for this study is clear and the introduction discussed the importance of the challenge in recycling polymers. The different characterization methods used in this study are clearly described and the results are well-presented. I believe more of such studies are needed for understanding the recycling challenges and to guide future directions in polymer recycling.
I recommend for publication after making one minor change. In Figure 1, 'mechanical properties' was repeated in which properties were investigated. This should be corrected.

Author Response

(The authors gave the same response as above.)

Reviewer 4 Report

Comments and Suggestions for Authors

Overall Comments

The authors have analyzed the possibility of recycling PP to fabricate glass fiber reinforced PP composites. Various values of recyclate fillers was used to analyze and establish the ideal value for manufacturing polypropylene composites with 30% glass fiber content, having satisfactory properties. However, the authors need to address the following comments on the technical aspects and English grammar.

Specific questions

1.     The Abstract of the manuscript introduces the work as something that can mitigate the two serious threats related to the future and our planet. However, the work involves recycling PP with glass fiber composites. How is using glass fiber going to help mitigate those threats. PP is recyclable, everyone knows that, but using glass fiber is increasing the carbon footprint of the world. The authors are more than welcome to work in this field, but claiming these huge aspects of mitigating the threats is hypocritical in this aspect. The authors are requested to introduce the Abstract and work in a different way. Probably, talking about only recycling PP (which is nothing new) is enough. The composite increases landfill issue, because these glass fiber reinforced PP composites will be ending up in landfill again, unless higher energy is consumed to extract and recycle the composites again. Using synthetic fibers and petroleum based polymers, and then talking about threats to the environment do not go hand in hand. These composites will increase that threat to the environment. Please change the Abstract and introduce the work in a more factual way.

2.     The authors have stated in Line 37-39 that, “Images showing, for example, the Great Pacific Garbage Patch, which is visible from Earth's orbit, have and should have a significant impact on the issue of reusing plastic waste [7,9,11-14].” However, no such image is visible in the manuscript. If the authors are talking about images in one of the references sited, then the authors need to rewrite the part as it is very confusing.

3.     The Introduction does the job of introducing the topic and illustrating some studies on recycling polymers. However, it fails to highlight the objective of the current work, the significance, novelty and basic details. The literature should find the gap, show similar studies and then highlight the main work at the end, stating how this work is going to impart some new knowledge. It should also have a few sentences that details a gist of the work carried out, why and how. All these aspects need to be added to the last paragraph of the Introduction.

4.     The authors have used % signs but have not specified if it is wt.% or any other form of % calculation that has been carried out. If wt.% then the authors are requested to mention the same every time in the manuscript. So, replace % with wt.%.

5.     The authors mention the recyclate was varied between 20% to 100%. The authors have failed to provide whether it is the percentage wt. of the composite, or the 70 wt.% PP used. This is very confusing, and the authors are requested to use percentage wt. of the composite, where if 30 wt.% of glass fiber is constant, the recyclate can be a maximum of 70 wt.%.

6.     The authors have failed to explain material used properly. The English is very confusing. What is recyclate and what is fillers? The authors need to explain. The authors need to use the same word every time while referring to the same thing. If recyclate and fillers are different, then what were the fillers. The section is very confusing and needs proper explanation with detailed segregation.

7.     The authors mentioned, “The recyclate was sourced from the pre-consumer waste stream.” However, this is not enough. The whole purpose of a publication is to ensure that this is repeatable by anyone anywhere in the world. The source needs to be detailed properly and the method of extraction or processing should also be detailed. If acquired from an industry, this needs to be mentioned as well. If processed in the lab, then the entire process should be detailed.

8.     Figure 1 needs better illustration and modification. The links are not established correctly. In the current form, it is not adding anything to the manuscript.

9.     The authors have mentioned fillers in the initial part of the manuscript, giving the impression that a particular filler has also been used to enhance the composite properties. However, Section 2, and specifically section 2.1 states otherwise. What are these fillers? Again, very confusing writing and using different words to refer to the same thing. Please rectify and explain. If there are no fillers, remove the term. Fillers are generally used to enhance certain performance of the composite.

10.  The authors need to provide reference or justification behind using the specified temperatures during extrusion process, and processing parameters for injection molding.

11.  Section 2.4 needs renaming. The current name for a subheading is too long.

12.  Mechanical properties have been mentioned, but only tensile has been studied. The authors are requested to perform bending studies on the specimens as well. Also, tensile is only a small part of mechanical properties. So, either mention tensile properties or perform other mechanical tests. Impact and hardness properties also fall under mechanical properties of any specimen. The difference is static loading and dynamic loading. The authors need to specify that. Bending is critical, and the authors are requested to perform bending analyses if possible or provide a valid justification to show why it is not needed.

13.  The authors need to prove whether the variation in densities of the samples is statistically significant. These are very small variations, with the largest variation being 0.05 g/cm3. These variations might not even be statistically significant. Also, using 2 points after decimal is more than enough in figure 3. Furthermore, the standard deviations or covariance of the results are also not shown. The average values should have standard deviations and covariance, which needs to be illustrated in the manuscript.

14.  Figure 4 needs to show Modulus of Elasticity as well for all the samples. Also, please use 2 points after decimal only. Standard deviations and/or covariance also needs to be illustrated in the figures and text where values are mentioned.

15.  The authors mention in Line 292, in section 3.1, that “The comparison of the obtained results is presented in Figures 4 and 5.” However, Figure 5 shows ass flow rate. Please rectify.

16.  There is an interesting result where the PPH^ GF30-reg has higher strength than PPH6 GF30-40/30. Also, the elongation decreases with decreasing recyclate, but then suddenly increases without any recyclate. These aspects need further clarification. The first thing is to understand whether these variations are statistically significant. Then, the reason behind the same. Some type of explanation is needed to justify, based on literature or other understanding.

17.  Discussion of elastic modulus is missing in section 3.2. Please add and discuss.

18.  Section 3.3 details only the result that can be clearly seen from the figure. But the authors have failed to discuss the reasons behind the deterioration. The authors are requested to use scientific understanding to explain why these changes are shown and what do these changes signify.

19.  The reason drawn behind variation in impact properties include melt mass flow index. However, the melt mass flow study shows lowest value for the PPH6 GF30-reg, a trend not achieved during impact analysis. Therefore, there are certainly other factors influencing the properties here. The authors need to provide a better explanation and understanding of the same.

20.  Again, no reason provided behind the PPH6 GF30-reg having lowest hardness. The authors are requested to provide reasons with scientific justification and not just mention results. This is a manuscript, not a report. Here the elastic modulus and flexural properties can have influence, which the authors have not even mentioned.

21.  The authors initially mentioned that the recyclate was used between 20% and 70%, however, the proportions used were only 20%, 30% and 70%, why were the intermediate values ignored? The authors need to justify the selection of recyclate amounts.

22.  The conclusion needs to be more detailed with numbers and values, rather than just stating “The obtained results also permit us to conclude that from the standpoint of the derived mechanical characteristics, impact strength, and deformability of the final product, it is advantageous to produce details from regranulate rather than producing polymer mixtures comprising the original material and recyclate”. Quantification of these aspects with numbers or percentage variation is required. The conclusion needs to be more detailed with results.

The paper needs more work, and all the comments should be addressed. There are some interesting studies in the manuscript, however the authors need to explain the reasons behind these observations, rather than just detailing the results. The Discussion part of the results is very limited and needs elaboration. The authors need to answer all the comments before the paper can be accepted for publication.

Comments on the Quality of English Language

English Language Comments

1.     Line 35-36 states, “Their great advantage is the possibility of reprocessing, which we often forget and allow us to waste them by recklessly throwing them away and storing them in landfills.” There are a lot of grammatical errors in this statement. The correct one will be, “Their great advantage is the possibility of reprocessing, which people often overlook, resulting in these ending up as landfills”.

2.     The authors state in Lines 65-67, “In the publication The Influence of Mechanical Recycling on Properties in Injection Molding of Fiber-Reinforced Polypropylene, presented in the International Polymer Processing magazine (2019), Evens et al. analyzed the…” A scientific way of writing this is, “Evens et al. (2019) analyzed the…” There is no need to mention the title of the paper. Please follow the convention of scientific presentation.

3.     Similar think repeated in Lines 77-79, 93-95 and 106-19. There is no need to mention titles and journal names, only author date reference is enough. The authors are requested to refer to high quality journals to understand how in-text citation is carried out, especially while mentioning the author details in the text of the manuscript. Often journals also have referencing guidelines. Please refer to the same for Materials or any other high impact journal from Springer or Elsevier.

4.     The Materials and Methods section has one paragraph per sentence. This is not acceptable. Please merge sentences into one paragraph and make paragraphs only when necessary (a paragraph is becoming too long, or drastically different aspects are being discussed)

5.     Figure 1 says, “hardenss”. Please rectify.

6.     The authors have mentioned “124 oC” in Line 388, with similar discrepancies in temperature values in other places. Please be consistent and use 124ºC in all places.

7.     The language needs proof-reading throughout the paper.

8.     ‘Regranulate’ is not an English word, ‘re-granulate’ can be used instead.

Author Response

(The authors gave the same response as above.)

Round 2

Reviewer 1 Report

Comments and Suggestions for Authors

Agree to accept the manuscript.

Comments on the Quality of English Language

Minor editing of English language required

Author Response

Dear Reviewer

First of all, I would like to thank you very much for your work in preparing the review and for all the comments that contributed to improving the quality of the manuscript and the way of presenting knowledge.

Best regards

Tomasz Stachowiak

Reviewer 2 Report

Comments and Suggestions for Authors

The paper is accepted in its current form.

Author Response

(The authors gave the same response as above.)

Reviewer 4 Report

Comments and Suggestions for Authors

The authors have satisfactorily addressed all the comments. However, some small errors are still present.

1. Line 17 - "...analyzes..." Here it should be "analyses".

2. The authors have removed Fig. 1, but have not corrected the figure numbers. The first figure now is Fig. 2.

3. The authors have mentioned that the parameters for extrusion and injection molding processes have been taken from literature and years of experience. The authors are therefore requested to provide references to the studies that have been referred to establish the parameters.

4. Answer to Question 21, "Additional explanations have been added regarding the mixtures produced (amounts of re-granulate used). The amounts of re-granulate used were dictated by the minimum noticeable re-granulate value in the case of the 50/20 composition, while the 40/30 composition was the maximum re-granulate value that could be used without a significant change and impact on the properties of the finished product, as shown by the tests of the melt flow rate index along with the increasing share of re-granulate deterioration processing properties are affected (the melt flow rate decreases significantly). Further increasing the amount of re-granulate could significantly worsen the processability of the composition. Of course, these are, to some extent, predictions of the behavior of this type of polymer composition. Due to the variability of the properties of polymeric materials, further work and research in this area is required", should be incorporated into the manuscript.

Comments on the Quality of English Language

Please proofread once to ensure minimal mistakes in the English Language.

Author Response

(The authors gave the same response as above.)
